# Groupwise registration of infant brain diffusion tensor images using intermediate subgroup templates

**Kuaikuai Duan**[1,2,3]*, **Longchuan Li**[1,2], **Vince D. Calhoun**[3], **Sarah Shultz**[1,2]*

**1** Marcus Autism Center, Children's Healthcare of Atlanta, Atlanta, Georgia, United States of America,
**2** Emory University School of Medicine, Department of Pediatrics, Atlanta, Georgia, United States of America, **3** Tri-Institutional Center for Translational Research in Neuroimaging and Data Science (TReNDS), Georgia State University, Georgia Institute of Technology and Emory University, Atlanta, Georgia, United States of America

* kuaikuai.duan@emory.edu (KD); sarah.shultz@emory.edu (SS)

## Abstract

Registering infant brain images is challenging, as the infant brain undergoes rapid changes in size, shape and tissue contrast in the first months of life. Diffusion tensor images (DTI) have relatively consistent tissue properties over the course of infancy compared to commonly used T1 or T2-weighted images, presenting great potential for infant brain registration. Moreover, groupwise registration using intermediate templates can reduce deformation and bias introduced by predefined atlases, but most methods use scalar (e.g., fractional anisotropy) images, which lack the microstructural orientation information in tensor images that can help differentiate brain structures and further improve infant image registration accuracy. Here, we propose an intermediate subgroup tensor template-based groupwise (IST-G tensor) registration approach to align infant tensor images to a sample-specific common space. First, tensor images are clustered into more homogenous subgroups using Louvain clustering based on image similarity. Within each subgroup, tensor images are aligned using DTI-toolkit to generate subgroup tensor templates, which are subsequently aligned to a sample-specific common space. Results show that our approach significantly improved registration accuracy both globally and locally compared to standard tensor-based and fractional anisotropy-based approaches. Clustering based on image similarity yielded significantly higher registration accuracy than no clustering and performed comparably to clustering by chronological age. By leveraging the consistency of features in tensor maps across early infancy and reducing deformation through intermediate subgroup tensor templates, our IST-G tensor registration framework facilitates more accurate alignment of longitudinal infant brain tensor images.

**Data availability statement:** All scripts are available in the public repository: (https://github.com/Luckykathy6/groupwiseRegister). The DTI data has been deposited in NeuroVault (https://identifiers.org/neurovault.collection:20732).

**Funding:** The work was funded by the National Institutes of Mental Health, USA (K01MH108741, 2P50MH100029, and R01MH119251 to SS); the National Institute of Biomedical Imaging and Bioengineering, USA (R01EB027147 to SS and VC); and funds from the Whitehead and Marcus Foundations (to SS). The sponsors did not play a role in the study design, data collection and analysis, decision to publish, or preparation of the manuscript.

## Introduction

Brain image registration—the alignment of individual brain images to a standard brain image (i.e., template)—is important for establishing spatial correspondence and facilitating group-level inferences [1,2]. A number of approaches have been proposed for registering brain images, such as FMRIB's Linear Image Registration Tool (FLIRT) [3], FMRIB's nonlinear image registration tool (FNIRT) [4], Symmetric Normalization (SyN) algorithm [5], Diffeomorphic Anatomical Registration Through Exponentiated Lie Algebra (DARTEL) [6] and its predecessor, Unified Segmentation [7] which is implemented in the Statistical Parametric Mapping (SPM) [8]. While these algorithms have been successfully and routinely applied to register adult brain images, registering longitudinal infant brain images presents unique challenges [9,10]. Over the course of infancy, the brain undergoes dramatic changes in size, morphology, myelination, and function [10–14], with significant changes occurring in the infant's brain almost every week [15]. Of particular relevance to registration, the relative signal intensities of gray and white matter in anatomical T1- and T2-weighted MRI images (the imaging modalities that are most commonly used for infant brain image registration [16–18]) reverse over the course of the first postnatal months [19–21] (see Fig S1 in S1 File). Given these rapid and substantial changes in tissue contrast and brain shape, infant brain images vary tremendously over developmental time, making it challenging to accurately identify and align corresponding brain features at different developmental stages.

A related challenge is the difficulty associated with selecting a template that is representative of the developmental variability within a longitudinal infant sample [22]. Selection of a representative template is critical in longitudinal studies because templates with features that are not well matched to the sample can introduce unnecessary deformations that may bias results [9,23]. Although several age-specific pediatric templates have been created [10,24,25], biases may still be introduced if the age-specific template is not closely matched to or equally representative of the age range under investigation [26,27]. For instance, after creating an age-specific template for a relatively narrow age range (39- to 42-weeks gestational age), Kazemi et al. demonstrated that even narrower age range templates (39–40 and 41–42 weeks) improved registration performance [28]. Given the fast pace of brain development in early infancy (with significant changes occurring on the order of days and weeks [15]) and the fact that individual infants develop on different time scales (with brain maturation unfolding more rapidly in some infants than others), registering infant images towards a sample-specific common space that is optimally representative of and specific to the sample of interest may yield more accurate registration than registering infant images to a predefined age-specific template (which may not be equally representative of all ages under investigation) [28].

Diffusion tensor images offer great advantages for aligning infant brain images compared to T1 or T2-weighted images (which are most commonly used for infant brain image registration), because DTI provides more stable tissue properties over development [29–32], and (unlike scalar maps) captures microstructural orientation of white matter tracts which can be used to further differentiate brain structures(see Fig

S1 in S1 File). A widely-used standard approach for DTI-based registration is DTI-ToolKit (DTI-TK, https://dti-tk.source-forge.net/) [33], which has been shown to outperform existing popular scalar image (e.g., fractional anisotropy or FA) based registration methods [34,35] in aging populations [33,36] and neonates with infantile Krabbe disease [35]. However, its performance in typically developing infants—particularly during periods of rapid brain development—remains underexamined. Additionally, DTI-TK has not yet been implemented within an intermediate template guided groupwise registration framework, which is a crucial consideration for optimizing infant brain images registration, as discussed below.

Groupwise registration enhances registration accuracy by simultaneously aligning all images to a common space (i.e., the average image) [37–40]. However, when inter-subject variation is large (e.g., in infants or large population data), using groupwise registration to align all images to a single data center (i.e., the average image) may not fully preserve the distribution of all population data [41]. To overcome this limitation, researchers have developed intermediate template-guided groupwise registration approaches for both brain structural [41,42] and diffusion tensor images [43,44], where brain images are first registered to intermediate templates and then progressively registered to the final template that is most representative of the sample (i.e., the final common space). This progressive registration strategy has been shown to minimize deformation—a major consideration in infant neuroimage studies [43,45]. Despite its advantages, most existing intermediate template guided groupwise registration approaches for DTI mainly utilized DTI derived scalar (FA or mean diffusivity (MD)) images [43,44], missing the inherent rich white matter microstructural orientation information contained in tensor images that can further enhance registration accuracy, especially for infants.

To address the challenges associated with aligning highly heterogeneous longitudinal infant images, we propose intermediate subgroup tensor template-based groupwise (IST-G tensor) registration, an approach that combines the developmentally stable tissue properties and abundant white matter microstructural information within DTI with the benefits of groupwise registration to a sample-specific common space. Unlike the standard tensor-based registration approach (DTI-TK), where all tensor images are registered together at one level to generate the sample-specific common space (Fig 1A), our approach registers tensor maps at two levels, first registering a group of DTI scans to an intermediate subgroup common space (within subgroup alignment) and then registering all resulting images to a sample-specific common space

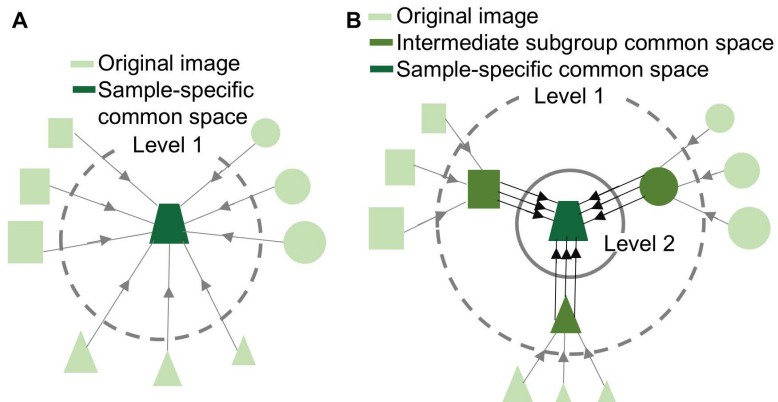

**Fig 1. Illustration of (A) standard registration and (B) the proposed IST-G tensor registration framework.** In standard registration (one level), all original (tensor or FA) images are directly aligned to a sample-specific common space. In the proposed IST-G tensor registration framework, original tensor images are first clustered into subgroups based on shared image characteristics. In the first level, tensor images within each subgroup are registered to their intermediate subgroup common space. In the second level, images that were aligned to the intermediate subgroup common space in the first level are further aligned to a sample-specific common space. Different shapes represent different subgroups formed on the basis of shared characteristics (chronological age or image similarity). The dashed circle represents level 1 registration, and the solid circle represents level 2 registration. Light green represents original tensor images, green represents the intermediate subgroup common space, and dark green represents the final sample-specific common space.

(across subgroup alignment, Fig 1B). The one level alignment only applies a single global alignment, while the two-level alignment incorporates both local (within subgroup) and global (across subgroup) alignment steps to better handle population heterogeneity.

We developed this registration approach using longitudinal infant DTI data collected from birth to 7 months, the most dynamic period of postnatal brain growth, providing a rigorous test case for evaluating our approach. Our aims were to: 1) replicate findings from aging populations demonstrating that standard tensor-based registration (DTI-TK) [33] outperforms scalar (FA)-based registration in infants; 2) compare the performance of the proposed IST-G tensor registration with standard tensor-based registration (DTI-TK)[33]; and 3) confirm that IST-G tensor registration(Fig 1B) generates smaller deformations compared to standard tensor-based registration (Fig 1A). We predicted that the proposed IST-G tensor registration will achieve higher accuracy compared to standard tensor-based registration and standard fractional anisotropy-based approaches. Finally, we also compared the impact of 3 different approaches for clustering images into subgroups—clustering based on chronological age (the predominant approach), clustering based on image similarity (which may yield more homogeneous subgroups, especially during periods characterized by rapid developmental change and/or individual differences in developmental timing), and no clustering—on registration accuracy. We predicted that clustering based on image similarity will yield higher registration accuracy compared to no clustering and compared to clustering by chronological age.

## Materials and methods

### Participants

This study included 27 typically developing infants (assigned sex at birth: 19 male and 8 female) who participated in a prospective longitudinal study [46] at the Marcus Autism Center, in Atlanta, GA, USA. Infants had a mean gestational age at birth of 39.09 weeks (SD = 1.40 weeks) and were considered to be typically developing based on their family and medical histories. Specifically, there was no family history of autism within up to 3rd degree relatives, no developmental delays in first-degree relatives, and no medical history of pre- or perinatal complications, seizures, known medical conditions or genetic disorders, hearing loss or visual impairments. Each infant was scanned at up to three timepoints between birth and 7 months, yielding a total of 53 diffusion MRI scans. The age distribution of scans is presented in Fig S2 in S1 File. The research protocol was approved by the Emory University Institutional Review board and written informed consent was obtained from the legal guardians of all infants.

### Diffusion MRI data acquisition

All infants were scanned at the Center for Systems Imaging Core at Emory University School of Medicine using a 3T Siemens Trio Tim system with a 32-channel head coil. All infants were scanned during natural sleep. First, infants were swaddled, rocked, and/or fed to promote natural sleep. Once asleep, the infant was placed on a pediatric scanner bed. Measures were taken to minimize scanner noise (e.g., < 80 dBA), including the use of pediatric sound-attenuating headphones with MR-safe optical microphones for real-time sound level monitoring and a custom acoustic hood inserted into the MRI bore [47]. White noise was gradually introduced through the headphones before the first scan sequence to help mask the onset of scanner noise. A MRI-compatible camera (MRC Systems) was mounted on the head coil to allow continuous visual monitoring of the infant during the scan. A trained experimenter stayed in the scanner room to oversee the scan session. The scan was stopped if the infant woke or if increased sound levels were detected.

Diffusion MRI data were collected using a multiband imaging sequence [48,49] with the following parameters: TR/TE = 6200/74 ms, multiband factor = 2, GRAPPA factor = 2, field of view = 184 × 184, matrix = 92 × 92, b-values = 0/700 s/mm$^2$, spatial resolution = 2 mm isotropic, 61 diffusion directions, 67 slices covering the entire brain, and extra six averages of b0 images (to enhance the signal-to-noise ratio of the baseline diffusion MRI signal). The diffusion MRI sequence had a total

scan time of 7 minutes 26 seconds. Additionally, a b0 image with the reverse phase encoding (posterior-to-anterior) was acquired to correct for susceptibility-induced distortion in diffusion MRI images [50,51].

## Diffusion MRI data preprocessing

Diffusion MRI data were preprocessed using FSL 5.0.9 and in-house Matlab code (Matlab 2023). Preprocessing steps included correction for eddy-current distortion and removal of susceptibility distortion using the *eddy* and *topup* functions in FSL [52,53]. Tensor maps and tensor-derived scalar maps, including maps of fractional anisotropy (FA) and mean diffusivity (MD), were calculated using FSL's function *dtifit* with weighted least-square tensor fitting. Weighted least-square fitting was used to minimize the impact of motion on the infant data [54].

## Diffusion MRI data registration

Three different registration approaches—standard FA-based registration (i.e., standard registration of scalar FA images), standard tensor-based registration, and the proposed IST-G tensor registration approach—were used and are described below.

## Standard registration of scalar FA images

FSL's linear registration tool "FLIRT" and deformable registration tool "FNIRT" were used to align infant FA scalar images to a sample-specific common space (see flowchart in Fig 2A). FNIRT is a medium-resolution nonlinear registration algorithm that has been previously used in developmental neuroimaging studies [55–57]. The iterative registration approach for aligning all scans onto a sample-specific common space (Fig 1A) is similar to the approach implemented in DTI-TK. Note that for each step of the registration, a single iteration was used as most studies do not employ multiple iterations for linear and non-linear registration for sample-specific scalar templates [25,28,58].

## Standard tensor-based registration (DTI-TK)

Infants' tensor maps were registered using the standard routine (see flowchart in Fig 2B, https://dti-tk.sourceforge.net/pmwiki/pmwiki.php%3Fn%3DDocumentation.FirstRegistration) in DTI-TK [33,36]. All participants' tensor maps were first aligned to an initial target tensor template (generation of the initial target tensor template is described below) using a 6-degree of freedom (dof) rigid body transformation (implemented as 'dti_rigid_reg' command in the DTI-TK). Aligned images from all participants were then averaged to generate a 6-dof rigid body intermediate tensor template. This process was then repeated by aligning all participants' tensor maps iteratively to the above-generated 6-dof rigid body intermediate tensor template via 12-dof affine transformations (implemented as 'dti_affine_reg' command in the DTI-TK). These aligned images were then averaged to create a 12-dof affine intermediate tensor template. Lastly, all participants' tensor maps were iteratively registered to the 12-dof affine intermediate tensor template using diffeomorphic transformation (via piecewise affine transformation that divides each image domain into uniform regions and transforms each region affinely, implemented as 'dti_diffeomorphic_reg' command in the DTI-TK) to generate the sample-specific common space.

## Generation of the initial target tensor template

The initial target tensor template was generated by applying the abovementioned standard tensor-based registration method to align a subset of DTI scans (37 scans ranging from 0–7 months). A tensor map of an infant with relatively clear tissue contrast was chosen as the tensor template for 6-dof rigid body transformation (selection of tensor template did not affect the shape and size of the resulting initial target tensor template, see S1 File for details on **effect of choosing different images for generating the initial target template**). This selected tensor template was nudged to closely match the origin of MNI space (the anterior commissure) and to be as straight as possible. Standard DTI-TK processing,

**Fig 2. Flowchart summarizing methodology for (A) standard FA-based registration, (B) standard tensor-based registration and (C) the proposed IST-G tensor registration.**

including six-dof rigid body transformation (via 'dti_rigid_reg' command), 12-dof affine transformation (via 'dti_affine_reg' command) and diffeomorphic transformation (via 'dti_diffeomorphic_reg' command), were applied as described above to obtain diffeomorphically transformed tensor maps, which were then averaged to create the initial target tensor template for standard registration of tensor images. For standard registration of scalar FA images, an FA map derived from the initial target tensor template was used as the initial target FA template.

## Groupwise registration of infant tensor brain images using intermediate subgroup templates

In the proposed IST-G tensor registration framework (see Fig 2C for flowchart), infant DTI scans were first clustered into more homogenous subgroups based on image similarity using Louvain clustering. DTI scans in each subgroup

were then aligned separately, as described in "**Standard tensor-based registration (DTI-TK)**". The resulting images from all subgroups were further aligned onto a sample-specific common space. These steps are described in detail below.

To cluster scans into homogenous groups, we first computed image similarity (i.e., similarity index). All participants' original tensor maps were aligned to the initial target tensor template using 6-dof rigid body transformation. FA and medial diffusivity (MD) maps were then derived from the resulting tensor maps (using "TVtool" in DTI-TK) and used to compute the distance between each FA and MD image pair using equations 1 and 2. These two DTI-derived metrics (FA and MD) were selected because FA maps differentiate between gray and white matter well, whereas MD maps show high contrast between brain tissue and cerebrospinal fluid [43]. Next, the pairwise distance of FA and MD maps were normalized to range between 0 and 1, and then used to compute the similarity index between image $A$ and $B$ using equation 3.

$$d\,(\boldsymbol{A},\boldsymbol{B})_{FA} = \sum_{i=1,\;j=1}^{m,n} (\boldsymbol{A}_{FA}(i,j) - \boldsymbol{B}_{FA}(i,j))^2 \tag{1}$$

$$d\,(\boldsymbol{A},\boldsymbol{B})_{MD} = \sum_{i=1,j=1}^{m,n} (\boldsymbol{A}_{MD}(i,j) - \boldsymbol{B}_{MD}(i,j))^2 \tag{2}$$

$$s\,(\boldsymbol{A},\boldsymbol{B}) = \frac{1}{\widetilde{d}(\boldsymbol{A},\boldsymbol{B})_{FA}} + \frac{1}{\widetilde{d}(\boldsymbol{A},\boldsymbol{B})_{MD}} \tag{3}$$

where $m,n$ are the total number of rows and columns in each image. $\boldsymbol{A}_{FA}(i,j)$ and $\boldsymbol{B}_{FA}(i,j)$ refer to the FA values of images $A$ and $B$ at $i$-th row and $j$-th column. $d\,(\boldsymbol{A},\boldsymbol{B})_{FA}$ is the distance of FA maps between images $A$ and $B$. $\boldsymbol{A}_{MD}(i,j)$ and $\boldsymbol{B}_{MD}(i,j)$ refer to the MD values of images $A$ and $B$ at $i$-th row and $j$-th column. $d\,(\boldsymbol{A},\boldsymbol{B})_{MD}$ is the distance of MD maps between images $A$ and $B$. $\widetilde{d}(\boldsymbol{A},\boldsymbol{B})_{FA}$ and $\widetilde{d}(\boldsymbol{A},\boldsymbol{B})_{MD}$ represent the normalized distance (values are between 0 and 1) of FA and MD maps between images $A$ and $B$. $s\,(\boldsymbol{A},\boldsymbol{B})$ is the similarity index between images $A$ and $B$.

We then applied Louvain clustering to the estimated similarity indices among all FA and MD maps to stratify scans into more homogeneous subgroups. Louvain clustering maximizes within-group connections and minimizes between-group connections [59,60]. The Louvain clustering was implemented in two iterative phases: modularity optimization and community aggregation. In the first phase, nodes are reassigned between communities to maximize the overall network modularity. In the second phase, communities are aggregated into single nodes, creating a new network for the next iteration. These steps are repeated until no further modularity gain (e.g., modularity gain < 1E-9) is achieved, resulting in a structure where nodes within the same community are more densely connected than nodes in different communities [60].

After the scans were clustered into more homogeneous subgroups based on their image similarity, level 1 registration was performed within each subgroup following steps described in "**Standard tensor-based registration (DTI-TK)**": the original tensor maps in each subgroup were aligned to their respective common space to generate intermediate subgroup tensor templates via 6-dof rigid body transformation, 12-dof affine and deformable transformations. In level 2 registration, the intermediate subgroup tensor templates from each subgroup were then aligned onto the sample-specific common space using standard tensor-based registration (Fig 1B). Finally, the original tensor maps were transformed from each individual's original space to the sample-specific common space via the transformations derived in the two-level alignment process.

### Effect of clustering strategy on IST-G tensor registration

To compare the effect of different clustering strategies on registration performance, we also considered (1) sub-grouping scans based on chronological age; and (2) no clustering: treating all scans as a single group and performing the IST-G tensor registration on this single group (i.e., 2-level registration without subgroups) for fair comparison with the proposed IST-G tensor approach (i.e., 2-level registration with subgroups). Registration performance following each clustering strategy was compared against that from Louvain clustering based on image similarity.

### Effect of brain masks and number of iterations on IST-G tensor registration

To evaluate whether the proposed IST-G tensor was robust to different brain masks, we compared registration performance when brain masks were selected with FA thresholds of 0.05 (i.e., whole-brain), 0.1 (white matter-enriched and some gray matter regions), and 0.25 (white matter heavy regions). Moreover, we examined the effect of varying the number of iterations in affine and deformable transformation stages.

### Robustness of the registration approaches

To evaluate the robustness of standard FA-based, standard tensor-based and the proposed IST-G tensor registration, we tested whether registration accuracy changes when randomly subsampling 50%, 60%, 70%, 80% and 90% of the full sample (n = 53 scans). Note for each random subsampling percentage, 10 random subsamples were drawn, and the same subsampling percentage was applied to each subgroup derived from Louvain clustering.

### Registration performance evaluation

We employed four commonly used metrics to evaluate the performance of different registration methods. The first metric is dyadic coherence, $\kappa$, which quantifies the variability in the aligned principal eigenvectors across scans [61,62]. Dyadic coherence ranges from 0 to 1, with 0 for randomly oriented tensor directions and 1 for perfectly aligned tensors (and highly aligned fibers) across scans [63]. The second metric is the voxel-wise normalized standard deviations across all FA ($\sigma_{FA}$) maps, which was computed for each voxel within the FA mask [36]. Suboptimal alignment strategies that overlap different white matter structures onto each other are expected to have high normalized standard deviation of FA. When plotting the empirical cumulative distribution functions (CDF), methods with better alignment are expected to have CDFs of $\kappa$ and $\sigma_{FA}$ to the right and left, respectively. Dyadic coherence and normalized standard deviation of FA can be computed using DTI-TK "TVMean" and "SVMean" commands, respectively. The third metric is the normalized mutual information (NMI) value [64] between each FA map of the aligned tensor map and the average FA map across all aligned tensor maps, which is computed by dividing their joint entropy by the sum of the marginal entropies using the Matlab package in [65]. NMI values range from 0 to 1 and reflect the similarity between each aligned FA map and their average. Larger NMI values indicate higher similarity (i.e., better registration) between each aligned scan and their average. The fourth metric is the Jacobian determinant of the deformation/transformation generated during registration, which is computed using the "CreateJacobianDeterminantImage" command in Advanced Normalization Tools (ANTs) and is evaluated both qualitatively and quantitively. Specifically, we computed and plotted the mean and standard deviation maps of the Jacobian determinant generated by each registration method. As Jacobian determinant values greater than 1 indicate expansion, less than 1 indicate compression, and 1 indicates no volume change, we computed how far (i.e., absolute difference values) the mean Jacobian determinant across the brain deviates from 1 for each registration method. Unless noted, all performance measures were computed for brain voxels with FA > 0.25 in the average FA map that was aligned to the sample-specific common space for fair comparison. Maps of $\sigma_{FA}$ were also generated and compared to evaluate the performance of the different registration methods.

## Statistical analysis

Registration accuracy—measured by NMI values and the absolute deviation of the mean Jacobian determinant from 1—was compared between each pair of methods (i.e., FA-based registration, standard tensor-based registration, and IST-G tensor) using two-sample, two-tailed t-tests. Effect sizes were quantified using Cohen's d and its 95% confidence interval, which were calculated with the MATLAB function "meanEffectSize.m". Bonferroni correction (for three pairwise comparisons) was applied to correct for multiple pairwise comparisons of registration accuracy.

## Results

### Standard FA-based registration vs. standard tensor-based registration

Figs 3A, 3C, 3D, 4A and 4B plot the performance of standard FA-based registration and standard tensor-based registration. Standard tensor-based registration generated smaller $\sigma_{FA}$ than FA-based registration (Fig 3A and the bottom row of Fig 3D) for both affine and deformable transformation stages, confirming previous findings in aging populations [33]. Moreover, standard tensor-based registration achieved significantly larger NMI values than FA-based registration (Fig 3C, $p < 1e-16$, Cohen's d (referred as d hereafter, 95% confidence interval (95% CI)) = 1.97 (1.63, 2.42), $t(52) = 54.27$), indicating that the similarity of FA maps derived from aligned tensor maps using standard tensor-based registration was significantly higher than that from standard FA-based registration. The mean and SD of FA maps from standard FA-based registration generated overall higher variability in gray matter than standard tensor-based registration, resulting in less well-defined gyri and sulci boundaries (Fig 3D). The absolute deviation of the mean Jacobian determinant from 1 generated by FA-based registration was not significantly different (i.e., no significant deformation difference) from that generated by standard tensor-based registration (Fig 4A, $p = 0.33$, d (95% CI)) = 0.09 (−0.41, 0.23), $t(52) = 0.98$), although qualitatively, FA-based registration generated slightly smaller mean and variability in terms of Jacobian determinant compared to standard tensor-based registration (Fig 4B). Overall, standard tensor-based registration outperformed standard FA-based registration by generating less variable and more similar FA maps.

### Standard tensor-based registration vs. IST-G tensor registration

In the proposed IST-G tensor, three subgroups (Fig S3A in S1 File) were identified using Louvain clustering based on image similarity with the default resolution value (i.e., resolution = 1). As expected, cluster membership was related to scan age and mean FA, but not driven by any single metric (Fig S3B-S3D in S1 File).

Compared to standard 1-level tensor-based registration, the proposed IST-G tensor (2-level) registration yielded smaller $\sigma_{FA}$ (Fig 3A and the bottom row of Fig 3D), smaller deformation as reflected by smaller absolute deviation of mean Jacobian determinant from 1 ($p = 5.66 \times 10^{-3}$, d (95% CI) = 0.42 (0.08, 0.79), $t(52) = 2.89$, Fig 4A) and smaller mean and variability in terms of Jacobian determinant (Fig 4B), larger dyadic coherence (Fig 3B), and significantly larger NMI values for FA maps (Fig. 3C, $p < 1E-16$, d (95% CI) = 2.91 (2.40, 3.55), $t(52) = 93.54$, i.e., significantly higher similarity between the aligned FA maps and their group average derived from all the aligned DTI scans). These differences in registration accuracy were observed in both affine and deformable alignment stages, but were particularly pronounced in the affine stage (Fig 3A and 3B), suggesting that the alignment to the subgroup specific space during the first level registration may be critical for the improved accuracy in the second level registration. In addition, the proposed IST-G tensor registration also generated significantly smaller deformation (Fig 4A and 4B, $p = 6.12 \times 10^{-3}$, d (95% CI) = 0.44 (0.13, 0.77), $t(52) = 2.86$) and larger NMI values for FA maps (Fig 3C, $p < 1E-16$, d (95% CI) = 4.92 (4.06, 6.00), $t(52) = 100.19$) than standard FA-based registration. A closer look at the maps of the standard deviation and $\sigma_{FA}$ generated using the three registration methods revealed that the splenium of the corpus callosum from standard FA-based registration yielded especially high $\sigma_{FA}$ values when compared to those from standard tensor-based or the proposed IST-G tensor) registration methods (Fig 3D, magenta circles). Examination of structures (Fig 3D, inset) in this region illustrates that differentiation between the

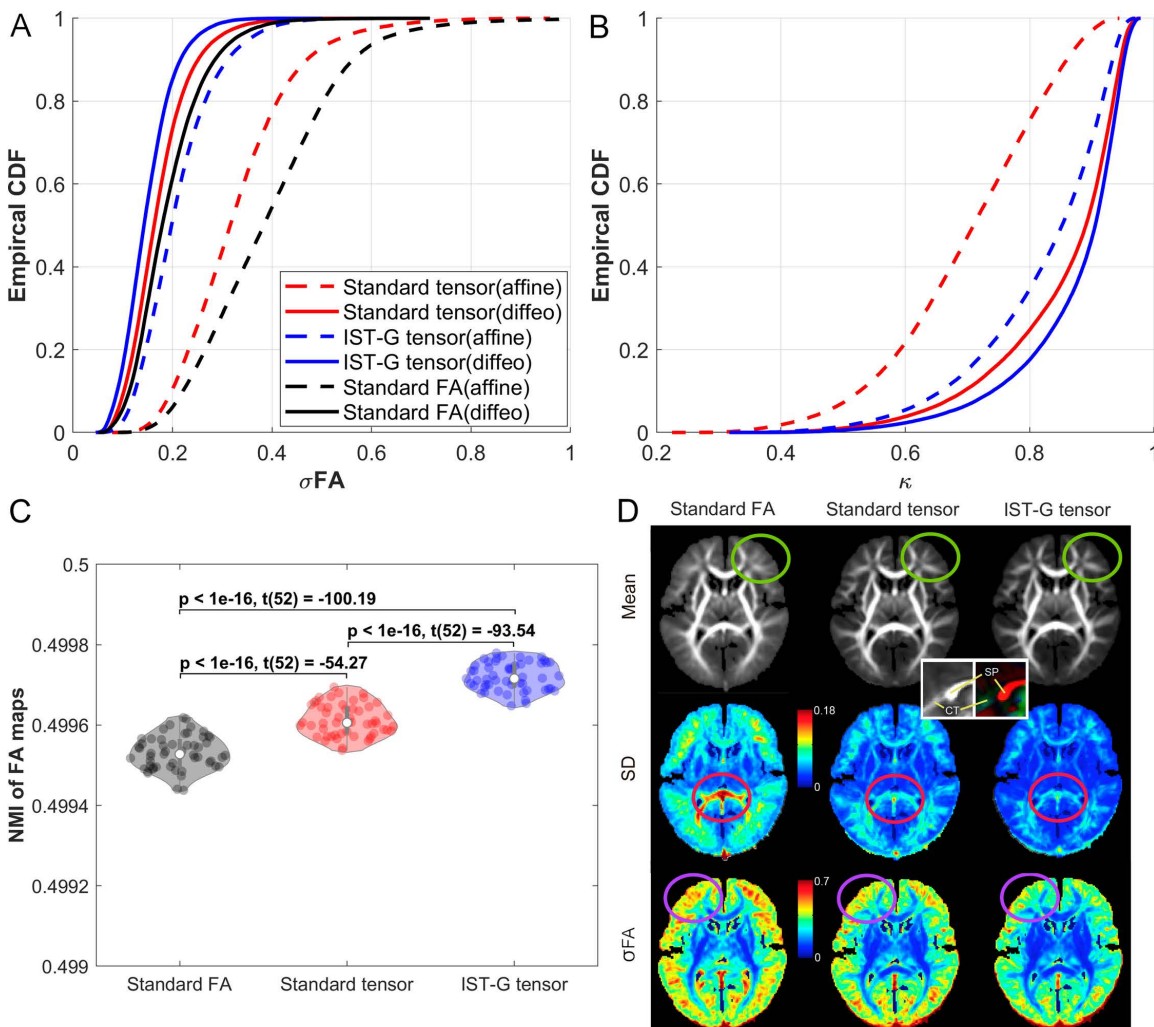

**Fig 3. Registration performance of standard FA-based, standard tensor-based, and the proposed IST-G tensor registration approaches. (A)** CDF plots of normalized standard deviation of FA maps ($\sigma_{FA}$) obtained from standard FA-based (black), standard tensor-based (red), and IST-G tensor (blue) approaches. For each of the registration methods, the dashed line represents results from the affine (linear) transformation stage and the solid line represents results from the diffeomorphic (diffeo, nonlinear) transformation stage (the same for subplot (B)). **(B)** CDF plots of dyadic coherence ($\kappa$) derived from standard tensor-based (red) and IST-G tensor (blue) approaches. Dyadic coherence was not evaluated for FA-based registration since tensor information was not used during FA-based registration. **(C)** NMI values and pairwise two-sample t-test statistics of FA maps derived from FA-based (black), standard tensor-based (red) and IST-G tensor (blue) approaches. **(D)** Mean, SD and $\sigma_{FA}$ of voxels within FA maps derived using each approach. Voxels within FA maps derived using standard FA-based registration show greater variability compared to standard tensor-based or IST-G tensor methods, especially at the splenium of the corpus callosum (magenta circles), leading to less well-defined gyri and sulci boundaries (green and purple circles). Overall, IST-G tensor registration generated the highest registration accuracy (as reflected by the smallest $\sigma_{FA}$, the largest dyadic coherence, and the largest NMI values), followed by standard tensor-based registration, and then standard FA-based registration.

splenium and the cerebellar tentorium—both of which have high FA values—can be achieved when using the distinct orientation information available in tensor, but not FA, maps.

### Effect of different clustering strategies on the proposed IST-G tensor registration

**Clustering by image similarity vs. by chronological age.** Longitudinal infant scans were clustered into 3 subgroups based on their chronological age: group 1: age<3 months (21 scans); group 2: age≥3 and age<6 months (25 scans);

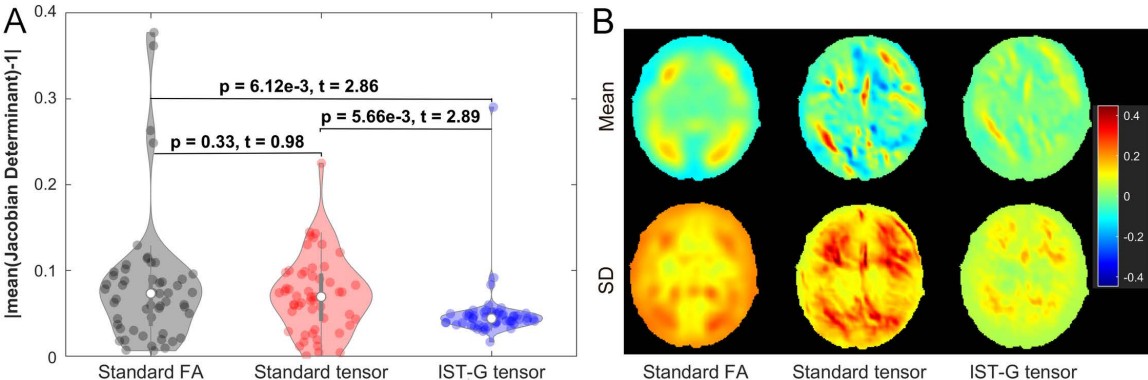

**Fig 4. Deformation generated by standard FA-based, standard tensor-based, and the proposed IST-G tensor registration approaches. (A)** The absolute deviations of mean Jacobian determinant from 1 and their pairwise two-sample t-test statistics for FA-based (black), standard tensor-based (red) and IST-G tensor (blue) approaches. **(B)** Mean and SD maps of Jacobian determinant generated by each registration approach. The proposed IST-G tensor registration generated the smallest deformation, followed by standard tensor-based registration, and then standard FA-based registration.

group 3: age ≥ 6 months (7 scans). For the proposed IST-G tensor registration, clustering by image similarity or by chronological age yielded comparable registration performance: CDF plots of dyadic coherence and $\sigma_{FA}$ are largely overlapping (Fig 5A and 5B) and MNI values of FA maps were not significantly different between clustering approaches (Fig 5C, $p = 0.24$, d (95% CI) = −0.02 (−0.05, 0.01), $t$ (52) = 1.19). This lack of difference may be explained by the relatively large amount of overlap between subgroups generated by each approach: 18 out of 26 scans (69%) in subgroup 1 clustered by image similarity were between 3 and 6 months of age; 17 out of 22 scans (77%) in subgroup 2 clustered by image similarity were between 0 and 3 months of age (Fig 5D).

**Clustering based on image similarity vs. no clustering.** Performing IST-G tensor registration without clustering (treating all 53 longitudinal infant DTI scans as a single group) yielded significantly lower registration accuracy at both affine and diffeomorphic transformation stages compared to clustering by image similarity (Fig 5(A)–5(C), $p = 4.77 \times 10^{-15}$, d (95% CI) = −2.10 (−1.73, −2.56), $t$ (52) = −10.90), indicating that the initial clustering of images into more homogeneous subgroups is critical.

**Clustering by image similarity or chronological age vs. no clustering.** Compared to no clustering, clustering by image similarity or by chronological age yielded significantly improved registration accuracy, as indexed by smaller $\sigma_{FA}$ and significantly larger NMI values (Fig 5(A)–5(C), clustering by chronological age vs. no clustering: $p = 1.01 \times 10^{-14}$, d (95% CI) = 2.11 (1.74, 2.57), $t$ (52) = 10.68). This indicates that the creation of relatively homogenous subgroups on the basis of shared features (image similarity or age) is critical for improving accuracy in groupwise tensor-based registration.

**Effect of varying brain masks and varying the number of iterations.** The proposed IST-G tensor registration outperformed standard tensor-based registration across a range of masking approaches (see S1 File for details on **effects of brain masks with different FA thresholds**). Moreover, increasing the number of iterations to register individual tensor images during the affine and diffeomorphic transformation stages increased registration accuracy, but the improvement was negligible when the number of iterations was greater than 2 (see S1 File for details on **effects of increasing number of iterations in the proposed IST-G tensor registration**).

**Robustness of the registration approaches.** Fig S6 in S1 File shows the box plot of the registration accuracy measured by NMI values and the absolute deviation of the mean Jacobian determinant from 1 when subsampling 50%, 60%, 70%, 80% and 90% of full samples. The proposed IST-G tensor registration always outperformed standard FA and standard tensor-based registration with larger NMI values and smaller deformation when increasing the subsampling percentage from 50% to 90%.

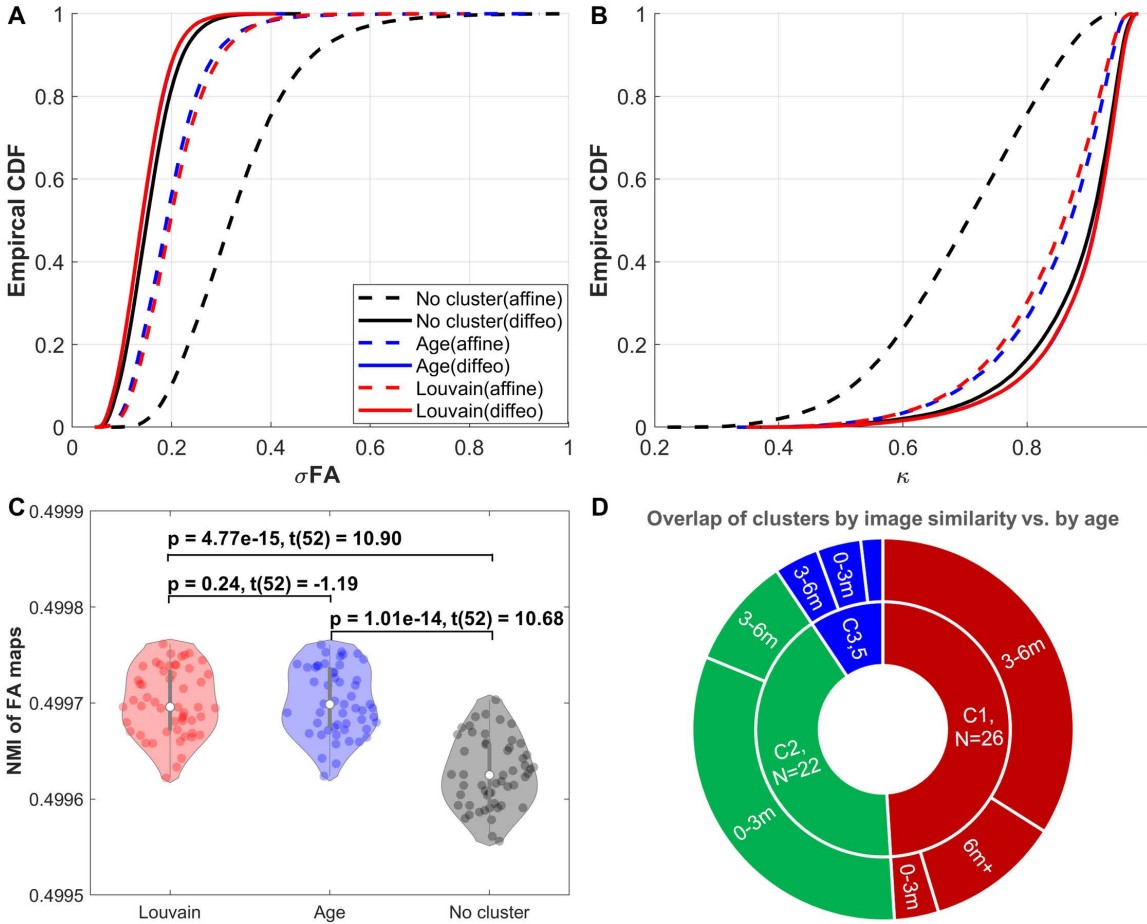

**Fig 5. CDF plots of (A) normalized standard deviations of FA (σFA) and (B) dyadic coherence (κ) derived from the proposed IST-G tensor registration clustered by image similarity (Louvain, red lines) or chronological age (Age, blue lines) or without clustering (No cluster, black lines).** For each of the clustering methods, dashed lines represent the results from the affine (linear) transformation stage and solid lines represent the results from the diffeomorphic (diffeo, nonlinear) transformation stage (the same for subplot (B)). (C) NMI values and pairwise two-sample t-test statistics of FA maps derived from the proposed IST-G tensor registration clustered by image similarity (Louvain, red) or chronological age (Age, blue) or without clustering (No cluster, black), and (D) Overlap between clusters generated by image similarity (C1-C3, the inner circle) and by chronological age (the outer circle, where 0-3m denotes age < 3 months, 3-6m represents age ≥ 3 and age < 6 months, and 6m+ denotes age ≥ 6 months).

## Discussion

Building on previous research demonstrating the advantages of tensor-based over scalar-based registration [33,36,66,67] and the benefits of intermediate template-based groupwise registration over standard registration [43,68,69], we developed an intermediate template-based groupwise registration framework for aligning infant brain tensor images collected between birth and 7 months, a period marked by very rapid postnatal brain growth and change. Briefly, infant DTI scans were first clustered into several smaller and homogenous subgroups based on image similarity using Louvain clustering. Then, standard tensor-based registration was implemented groupwise: first to align all tensor images within a subgroup to their intermediate subgroup common space, and then to register the images in the intermediate subgroup common space to the sample-specific common space.

Compared to scalar (FA)-based registration, both standard tensor-based and the proposed IST-G tensor registration improved registration accuracy globally (as quantified by smaller normalized standard deviations of FA, and larger NMI

values between aligned FA images and their average) and locally (as indicated by more sharply defined gyri and sulci boundaries, especially in the splenium of the corpus callosum), confirming that differentiation of distinct brain structures with similar anisotropic FA values can be achieved using the orientation information embedded in tensor maps [33,67].

Compared to both standard FA-based and standard tensor-based registration, the proposed IST-G tensor registration significantly improved registration accuracy as quantified by larger dyadic coherence of the principal eigenvector of the tensor maps, smaller normalized standard deviations of FA, larger NMI values between aligned FA images and their average, and smaller deformation. These improvements in registration accuracy were observed in both affine and deformable alignment stages, but were particularly pronounced in the affine stage, suggesting that first level registration of tensor images within subgroups may be critical for improved accuracy in the affine stage during second level registration.

Importantly, clustering of images into subgroups impacted the accuracy of IST-G tensor registration. Clustering based on image similarity and clustering based on chronological age outperformed no clustering, suggesting that creating more homogeneous subgroups with shared features (in this case, image similarity or chronological age) is critical for yielding improved registration performance. Contrary to our initial predictions, clustering based on image similarity did not improve registration accuracy compared to clustering based on chronological age, a null result that may be explained by the largely overlapping subgroups generated by each approach. Given that clustering by image similarity performs as well as clustering by chronological age, it may be advantageous to adopt the former approach when faced with uncertainty about which age cutoffs are most likely to yield homogeneous subgroups, or when working across developmental periods characterized by pronounced individual differences in developmental timing (with some infants maturing on different time scales than others). Finally, while the current study applied Louvain clustering to a normalized distance-based similarity measure—a popular and effective way to stratify scans into more homogenous subgroups—future research should explore more advanced image similarity measures (e.g., NMI [64] that captures both linear and nonlinear relationships and is less sensitive to outliers) and clustering methods (e.g., Leiden [70] algorithm that can capture hierarchical community structures).

While the present study focused primarily on registration of diffusion scans, our approach can also be useful for registering infant anatomical and functional images collected within the same session as diffusion scans. Specifically, the warping/deformation fields derived from the IST-G tensor registration of DTI scans can be readily applied to anatomical and functional images collected in the same session, potentially providing more accurate alignment for these imaging modalities especially during early infancy when gray and white matter tissue contrasts are isointense in T1- and T2-weighted images. Although we developed this registration approach using data collected from birth to 7 months—a particularly dynamic period of growth, providing a rigorous test case for this approach—this registration strategy can be readily applied to participants outside of our age range. For instance, this approach could advance studies focusing on pediatric populations that require more accurate brain maps, such those aimed at identifying abnormal brain regions associated with neurological conditions like epilepsy, potentially enhancing our understanding of brain connectivity and improving treatment strategies [71–73].

The findings presented in this study should be considered in context with its strengths and limitations. It should be noted while sample-specific templates have the advantage of minimizing deformations between individual images and the sample-specific common space, sample-specific templates usually lack standardized stereotaxic coordinates, making stereotaxic mapping and cross-study comparisons challenging [9]. A potential solution is to report findings based on brain regions (instead of coordinates) using standard parcellations [10,74]. If coordinate-based reporting of results is necessary, a transformation from the sample-specific template to standard stereotaxic space can be performed [10,24,75]. Additionally, this study uses Louvain clustering, which relies on modularity as its sole optimization criterion. This approach may miss meaningful structures in cases where modularity is not the best measure of community quality for the particular network. Like other clustering methods, Louvain clustering tends to be more susceptible to over-fitting in small datasets. Future studies should include more scans and explore clustering methods with other optimization criterion, such as spectral clustering [76,77].

As research increasingly focuses on mapping trajectories of brain development during infancy [78–81]—a highly dynamic period that likely exerts a strong influence on neurodevelopmental disorders [82]—there is a growing need for methodological tools designed to address the unique challenges inherent to longitudinal infant neuroimaging research. Here we present a novel IST-G tensor registration approach (made publicly available at https://github.com/Luckykathy6/groupwiseRegister), specifically designed to address challenges inherent to registration of rapidly changing infant brain tensor images. Given that accurate alignment of brain structures across participants is a cornerstone for atlas-based analyses of developing brains, we believe that the proposed method will aid in advancing understanding of early brain development, a critical imperative for supporting children with neurodevelopmental disorders.

## Supporting information

**S1 File. Supporting Methods, Supporting Results, Figure S1, Figure S2, Figure S3, Figure S4, Figure S5 and Figure S6.**
(DOCX)

## Acknowledgments

We greatly appreciate the families and their infants who participated in this study. We thank research coordinators, Brittney Sholar, Carly Reineri, Joanna Beugnon, and MRI Technician, Michael White, for their contributions to data collection. We extend our gratitude to Dr. Lei Zhou and Michael Valente for their support with equipment design and data acquisition protocols, Mahmoud Zeydabadinezhad for his assistance with data processing, and Zening Fu for providing valuable feedback on the manuscript.

## Author contributions

**Conceptualization:** Kuaikuai Duan, Longchuan Li, Sarah Shultz.

**Data curation:** Kuaikuai Duan, Longchuan Li.

**Formal analysis:** Kuaikuai Duan, Longchuan Li.

**Funding acquisition:** Longchuan Li, Vince D. Calhoun, Sarah Shultz.

**Investigation:** Kuaikuai Duan, Longchuan Li.

**Methodology:** Kuaikuai Duan, Longchuan Li.

**Software:** Kuaikuai Duan.

**Supervision:** Sarah Shultz.

**Validation:** Kuaikuai Duan.

**Visualization:** Kuaikuai Duan, Longchuan Li.

**Writing – original draft:** Kuaikuai Duan, Longchuan Li, Sarah Shultz.

**Writing – review & editing:** Kuaikuai Duan, Vince D. Calhoun, Sarah Shultz.

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
