## [Decision Letter · Decision Letter 0]

Dear Dr. Shultz,

Thank you for submitting your manuscript to PLOS ONE. After careful consideration, we feel that it has merit but does not fully meet PLOS ONE’s publication criteria as it currently stands. Therefore, we invite you to submit a revised version of the manuscript that addresses the points raised during the review process.

We look forward to receiving your revised manuscript.

Kind regards,

Federico Giove, PhD

Academic Editor

PLOS ONE

Journal Requirements:

Reviewers' comments:

Reviewer's Responses to Questions

**Comments to the Author**

1. Is the manuscript technically sound, and do the data support the conclusions?

Reviewer #1: Yes

Reviewer #2: Yes

2. Has the statistical analysis been performed appropriately and rigorously?

Reviewer #1: Yes

Reviewer #2: Yes

3. Have the authors made all data underlying the findings in their manuscript fully available?

Reviewer #1: Yes

Reviewer #2: Yes

4. Is the manuscript presented in an intelligible fashion and written in standard English?

Reviewer #1: Yes

Reviewer #2: Yes

Reviewer #1: In this piece of work, Kuaikuai et al., have explored the utility of using a new method based on subgroup tensor template-based groupwise registration to provide more accurate alignment of longitudinal infant brain tensor images.

From a cohort of 27 typically developing infants, the authors studied three different registration approaches (i.e., the standard tensor-based registration, the standard FA-based registration, and their groupwise tensor-based registration) on 37 DTI scans for testing their registration accuracy. They also compared the impact of such methods on the registration accuracy when clustering images into subgroups. Particularly, they used four different metrics, such as Dyadic coherence, voxel-wise normalized standard FA deviations, normalized mutual information value, and the Jacobian determinant of the deformation generated during registration, to evaluate the performance of these registration approaches. Finally, they explored how the proposed intermediate subgroup tensor template-based groupwise registration performed across different clustering methods.

Their findings suggest that both the standard tensor-based and intermediate subgroup tensor template-based groupwise registrations achieved better registration accuracy compared to the FA-based approach at both the global and local level, by providing smaller deviations of FA values and more defined gyri and sulci boundaries, respectively. They also observed that clustering by image similarity or by chronological age yielded comparable registration performance when using the intermediate subgroup tensor template-based groupwise registration. Moreover, they showed that performing intermediate subgroup tensor template-based groupwise registration without clustering (treating all infant scans as one group) resulted in lower registration accuracy at both affine and diffeomorphic transformation stages compared to clustering by image similarity. Finally, they found that their proposed method obtained an increased registration accuracy when varying either brain masks or the number of iterations during the affine and diffeomorphic transformation steps.

The goal of the study is well accepted in the field and the overall idea of the study is properly designed. The text is generally well written, and the addition of explanatory details facilitates the understanding of the study to a more generalized audience. However, the organization of some paragraphs (and sub-paragraphs) should be slightly improved; for example, the introduction could be written without dividing it in different paragraphs, but by blending some concepts in a unique “block”. It is also highly encouraged a major improvement of the figures (in which crucial information are completely missing); such effort has been nicely done for some of the figures in the supplementary material which contain more descriptive information than the main ones. Although the study is well developed, a few methodological questions and concepts need to be further explored:

• The “Registration of diffusion tensor images”, “Groupwise registration using intermediate templates”, and “Groupwise registration of infant tensor brain images using intermediate subgroup templates” paragraphs should be integrated inside the “Introduction” parts, without separating it from the rest of the speech. As it is right now, it is confusing for the reader.

• Figure 1 (both panel A and B) needs work. The authors should use an actual slice of a scan, instead of green squares, triangles, and circles, and somehow show different groups of such scans based on their characteristics. Also, what does the dashed circle represent? The first level? This should be specified. The sample-specific common space is poorly represented as well. I am sure the authors have actual images to better explain these concepts. In the caption, it is better to set in bold font the “Illustration of (A) standard registration and (B) the proposed intermediate subgroup tensor template-based groupwise registration framework” phrase that represents the figure’s title.

• Although the aims of the goal are well defined, the hypothesis (hypotheses) is (are) not stated. Thus, it is recommended to also add the hypotheses to help the readers understanding the overall purpose of the study and what the authors expect from the conducted analysis.

• The author used “DTI-TK Diffusion Tensor Imaging ToolKit” for the standard tensor-based registration. However, such software is only for Linux and Mac/OS and this could be a limitation for readers who do not have such operating systems but would like to replicate the methods. Did the author explored other software compatible for other operating systems to perform such analysis? If yes, were the results different?

• Which method was used to generate the 6-gof rigid body intermediate tensor? This information should be included in the “Standard tensor-based registration (DTI-TK)” paragraph for easy reproducibility of the study.

• It is unclear if the “Generation of the initial target tensor template” paragraph is part of the previous one [i.e., “Standard tensor-based registration (DTI-TK)”] or not. My understanding is that this paragraph is the explanation of how the initial target tensor template was generated. Which software was used to generate the initial target tensor template?

• It is highly encouraged to create a figure summarizing the methodology of the study; in particular, one that provide a simple overview of the steps performed for each registration approach.

• The “Groupwise registration of infant tensor brain images using intermediate subgroup templates” paragraph confuses the reader on the next steps performed in the analysis. The similarity index between images was computed using the FA and MD maps; how were the values extracted? How was the distance of these DTI-derived metrics between images computed? Were these indices used for the Louvain clustering approach?

• How was the Louvain clustering approach computed?

• The link for the standard routine registration did not work (https://dti175tk.sourceforge.net/pmwiki/pmwiki.php/Documentation.FirstRegistration/). Can the authors provide the correct one?

• It would be great if the authors can provide a more detailed descriptions on the differences between the one- and multiple-level alignment processes.

• What is the rational of performing two rounds of standard tensor-based registration on the scans for the “no clustering” analysis?

• Can the authors explain how were the four metrics to evaluate registration performance computed? Were they extracted using the DTI-TK software? If not, the authors should specify this information in the text.

• The quality of all main figures is poor. Can the author provide a better resolution? Also, it is recommended to avoid acronyms in the caption (or at least explain them first), allowing the reader to better interpret each figure.

• Although the authors somehow provided details, it is still difficult to do not get lost when reading the different terminologies (e.g., intermediate subgroup common space, sample-specific common space, groupwise registration, standard registration) used in the text. If possible, can the authors provide more insights for these terms to avoid possible confusion?

• The registration accuracy of all methods was performed on a total of 37 DTI scans. However, without considering other variants, this can improve when a higher number of scans are considered. Can the authors provide evidence that such results do not change when varying the number of scans in each method?

• It is highly encouraged to add a “Statistical Analysis” paragraph, summarizing all the tests performed to achieve the goals of the study. As of right now, this information is missing.

• The authors should at least mentioned future applications of their method in other fields. For example, several studies in the epilepsy field explored the role of functional connectivity to identify those regions whose resection leads to seizure freedom (see Goodale et al., 2020; Corona et al., 2023; Shah et al., 2019; Lagarde et al., 2018), but only few of them explored how structural connections can influence functional connectivity. Thus, it would be interesting to propose as future investigation whether such template can be extended to pediatric populations and be used to provide a more accurate brain maps, which can be essential for studying brain connectivity, identify abnormal regions, and ultimately developing better treatments for epilepsy.

Reviewer #2: The study by Shultz and colleagues addresses an important challenge in infant brain image registration: combining Louvain clustering with intermediate subgroup tensor templates, Shulz and collaborators approach offers a new solution tailored to the complexities of longitudinal infant brain image registration. This integration addresses the rapid developmental changes in infant brains, providing a more accurate and robust alignment method not previously reported in the literature. The method is logically and well presented as the clustering method and validation metrics. Crucially, the study compares multiple registration approaches (standard tensor-based, FA-based and intermediate subgroup registration), providing empirical support for the proposed method. The study introduces a novel framework for groupwise alignment of longitudinal infant DTI images by integrating Louvain clustering to create intermediate subgroup tensor templates, enhancing registration accuracy. Overall the manuscript is strong, but I have some considerations about it, I will completely support manuscript pubblication if they will be adressed:

1. which kind of correction for multiple comparisons is applied to the statistical comparisons? Author should better clarify this in the main text

2. Comparisons in the results are reported with p-values, but effect sizes are missing. Including effect sizes (Cohen’s d or standardized mean difference) would help quantify the significance of improvements. Since the p-values only indicate whether a difference is statistically significant. This will further clarify the strength of the effect beyond just statistical significance.

3. Instead of just reporting p-values, it would be better to provide 95% confidence intervals (CIs) for key statistics. CIs will help to interpret the precision of the effect size and whether the difference is practically meaningful.

4. The Discussion could better highlight the practical impact of this method in broader neuroimaging applications, beyond just registration accuracy improvements.

5. A more detailed discussion of limitations would strengthen the manuscript. Are there scenarios where this clustering method might not work well? Are there risks of overfitting to small datasets?

**Do you want your identity to be public for this peer review?** For information about this choice, including consent withdrawal, please see our Privacy Policy

Reviewer #1: No

Reviewer #2: No

---

## [Author Response · Author response to Decision Letter 1]

8 May 2025

Dear Reviewers,

We appreciate your time and effort in helping us to improve the manuscript. We have revised our manuscript entitled “Groupwise registration of infant tensor brain images using intermediate subgroup templates”, according to your constructive and valuable comments. In the following, we provide detailed responses (in blue color) to each comment. Corresponding changes are highlighted in our revised manuscript.

Reviewer #1: In this piece of work, Kuaikuai et al., have explored the utility of using a new method based on subgroup tensor template-based groupwise registration to provide more accurate alignment of longitudinal infant brain tensor images.

From a cohort of 27 typically developing infants, the authors studied three different registration approaches (i.e., the standard tensor-based registration, the standard FA-based registration, and their groupwise tensor-based registration) on 37 DTI scans for testing their registration accuracy. They also compared the impact of such methods on the registration accuracy when clustering images into subgroups. Particularly, they used four different metrics, such as Dyadic coherence, voxel-wise normalized standard FA deviations, normalized mutual information value, and the Jacobian determinant of the deformation generated during registration, to evaluate the performance of these registration approaches. Finally, they explored how the proposed intermediate subgroup tensor template-based groupwise registration performed across different clustering methods.

Their findings suggest that both the standard tensor-based and intermediate subgroup tensor template-based groupwise registrations achieved better registration accuracy compared to the FA-based approach at both the global and local level, by providing smaller deviations of FA values and more defined gyri and sulci boundaries, respectively. They also observed that clustering by image similarity or by chronological age yielded comparable registration performance when using the intermediate subgroup tensor template-based groupwise registration. Moreover, they showed that performing intermediate subgroup tensor template-based groupwise registration without clustering (treating all infant scans as one group) resulted in lower registration accuracy at both affine and diffeomorphic transformation stages compared to clustering by image similarity. Finally, they found that their proposed method obtained an increased registration accuracy when varying either brain masks or the number of iterations during the affine and diffeomorphic transformation steps.

The goal of the study is well accepted in the field and the overall idea of the study is properly designed. The text is generally well written, and the addition of explanatory details facilitates the understanding of the study to a more generalized audience. However, the organization of some paragraphs (and sub-paragraphs) should be slightly improved; for example, the introduction could be written without dividing it in different paragraphs, but by blending some concepts in a unique “block”. It is also highly encouraged a major improvement of the figures (in which crucial information are completely missing); such effort has been nicely done for some of the figures in the supplementary material which contain more descriptive information than the main ones. Although the study is well developed, a few methodological questions and concepts need to be further explored:

Response: We appreciate the time and effort invested by the Reviewer in providing these insightful comments. Responses to comments regarding paragraph organization and figures are provided in response to comments #1 and #2 below.

1. The “Registration of diffusion tensor images”, “Groupwise registration using intermediate templates”, and “Groupwise registration of infant tensor brain images using intermediate subgroup templates” paragraphs should be integrated inside the “Introduction” parts, without separating it from the rest of the speech. As it is right now, it is confusing for the reader.

Response: Thank you for this suggestion. We have revised the Introduction to integrate Registration of diffusion tensor images”, “Groupwise registration using intermediate templates”, and “Groupwise registration of infant tensor brain images using intermediate subgroup templates” paragraphs (see page 4-6, lines 76-122, highlighted).

2. Figure 1 (both panel A and B) needs work. The authors should use an actual slice of a scan, instead of green squares, triangles, and circles, and somehow show different groups of such scans based on their characteristics. Also, what does the dashed circle represent? The first level? This should be specified. The sample-specific common space is poorly represented as well. I amyh sure the authors have actual images to better explain these concepts. In the caption, it is better to set in bold font the “Illustration of (A) standard registration and (B) the proposed intermediate subgroup tensor template-based groupwise registration framework” phrase that represents the figure’s title.

Response: Thank you for this helpful comment. We originally used shapes to evoke the idea of clustering images into subgroups based on shared features (i.e. chronological age or image similarity). We attempted to use real brain images as you suggested, but we felt it was hard for the reader to identify shared features in the brain images with the naked eye (owing to relatively subtle differences in image features and their small size within the figure). Thus, while we appreciated your suggestion, we respectfully chose to retain the use of shapes to signify different subgroups.

To improve the clarity of the figure, we have now added additional detail to the figure caption to explain our intended use of different shapes, as well as the significance of the dashed circle and solid circle (denoting level 1 and level 2 registration, respectively). We also specified that the colors are used to denote the original tensor images (light green), intermediate subgroup common space (green), and the final sample-specific common space (dark green). Finally, we also bolded the figure title as you suggested (see Figure 1 caption, page 6, lines 123-132).

3. Although the aims of the goal are well defined, the hypothesis (hypotheses) is (are) not stated. Thus, it is recommended to also add the hypotheses to help the readers understanding the overall purpose of the study and what the authors expect from the conducted analysis.

Response: Thank you for the recommendation. We have added two hypotheses in the last paragraph of Introduction (see highlighted section on page 5, lines 115-116 and page 6, lines 120-122): (1) We predict groupwise registration using intermediate subgroup templates will achieve higher accuracy compared to standard tensor-based registration and standard fractional anisotropy-based approaches. (2) Clustering based on image similarity will achieve higher registration accuracy compared to no clustering and compared to clustering by chronological age.

4. The author used “DTI-TK Diffusion Tensor Imaging ToolKit” for the standard tensor-based registration. However, such software is only for Linux and Mac/OS and this could be a limitation for readers who do not have such operating systems but would like to replicate the methods. Did the author explored other software compatible for other operating systems to perform such analysis? If yes, were the results different?

Response: Thank you for this comment. Our groupwise registration using intermediate subgroup templates (IST-G) algorithm is built on DTI-TK, which is only available for Linux and Mac/OS. However, we found a workaround to call the binary executables of DTI-TK under Windows using Windows Subsystem for Linux (WSL). Therefore, it is recommended to use WSL to run IST-G tensor under Windows. Here are the instructions (we have incorporated this in the released package):

(1) Install WSL by following the instructions at https://learn.microsoft.com/en-us/windows/wsl/install

(2) IST-G tensor is being developed under Rocky Linux 9.5 (Blue Onyx), which is the suggested distribution.

(3) Once WSL is installed, open a Windows Powershell and type "bash". Now you are in Linux.

(4) All your windows drives are mounted under /mnt

(5) Follow the IST-G tensor instructions in the “READ ME.txt” in the released package to run it under WSL.

5. Which method was used to generate the 6-gof rigid body intermediate tensor? This information should be included in the “Standard tensor-based registration (DTI-TK)” paragraph for easy reproducibility of the study.

Response: Thank you for this comment. We used the 6-degree of freedom (dof) rigid body transformation (‘dti_rigid_reg’ command in the DTI-TK) to align all tensor maps to the initial target tensor template, and the aligned tensor maps were averaged to generate the 6 dof rigid body intermediate tensor template. We have included this information in the revised manuscript (see highlighted section, page 8, lines 185-189).

6. It is unclear if the “Generation of the initial target tensor template” paragraph is part of the previous one [i.e., “Standard tensor-based registration (DTI-TK)”] or not. My understanding is that this paragraph is the explanation of how the initial target tensor template was generated. Which software was used to generate the initial target tensor template?

Response: We apologize for the confusion. Yes, the “Generation of the initial target tensor template” paragraph explains how the initial target tensor template was generated. We used DTI-TK to generate the initial target tensor template. We have clarified this in the revised manuscript (see highlighted section, page 9, lines 204-206).

7. It is highly encouraged to create a figure summarizing the methodology of the study; in particular, one that provide a simple overview of the steps performed for each registration approach.

Response: Thank you for this important suggestion! We now include a flowchart for each registration approach in the revised manuscript (see Figure 2). The flowchart is also reprinted below.

Figure 2. Flowchart summarizing methodology for (A) standard FA-based registration, (B) standard tensor-based registration and (C) the proposed IST-G tensor registration.

8. The “Groupwise registration of infant tensor brain images using intermediate subgroup templates” paragraph confuses the reader on the next steps performed in the analysis. The similarity index between images was computed using the FA and MD maps; how were the values extracted? How was the distance of these DTI-derived metrics between images computed? Were these indices used for the Louvain clustering approach?

Response: We apologize for the confusion. The similarity index between tensor images was computed using the FA and MD maps, which were extracted using the DTI-TK “TVtool” command. The distance between each pair of FA and MD maps was computed using equations 1 and 2. The obtained similarity indices were used for Louvain clustering. We have added these details to the manuscript and clarified our approach (see highlighted section, page 10, lines 217-234).

9. How was the Louvain clustering approach computed?

Response: Thank you for this question. The Louvain clustering was implemented in two iterative phases: modularity optimization and community aggregation. In the first phase, nodes are reassigned between communities to maximize the overall network modularity. In the second phase, communities are aggregated into single nodes, creating a new network for the next iteration. These steps are repeated until no further modularity gain (e.g., modularity gain <1E-9) is achieved, resulting in a structure where nodes within the same community are more densely connected than nodes in different communities1. We now include this information in the revised manuscript (see highlighted section, pages 10-11, lines 236-241).

10. The link for the standard routine registration did not work. Can the authors provide the correct one? (https://dti175tk.sourceforge.net/pmwiki/pmwiki.php/Documentation.FirstRegistration/).

Response: We apologize for the incorrect link. We have updated the DTI-TK standard registration link (https://dti-tk.sourceforge.net/pmwiki/pmwiki.php%3Fn%3DDocumentation.FirstRegistration) in the revised manuscript (see highlighted section, page 8, lines 184-185).

11. It would be great if the authors can provide a more detailed descriptions on the differences between the one- and multiple-level alignment processes.

Response: Thank you for this suggestion. In the standard tensor-based registration (i.e., one-level alignment), each tensor image is directly registered to the initial target tensor template, and the aligned images are averaged voxel-wise to construct a sample-specific template. In the proposed groupwise registration using intermediate subgroup tensor templates (i.e., 2-level alignment), we employ the following steps: 1) cluster DTI scans into homogenous subgroups using Louvain clustering based on image similarity; 2) register each subgroup of homogenous DTI scans to their intermediate subgroup common space using standard tensor-based registration (within subgroup alignment), 3) register intermediate subgroup tensor templates from each subgroup to the sample-specific tensor template using standard tensor-based registration (across subgroup alignment). The one level alignment applies a single global alignment, while the two-level alignment incorporates both local (within subgroup) and global (across subgroup) alignment steps to better handle population heterogeneity. These descriptions are now included on page 5, lines 105-108.

12. What is the rational of performing two rounds of standard tensor-based registration on the scans for the “no clustering” analysis?

Response: Thank you for this important question. To ensure a fair comparison with the proposed IST-G tensor registration (i.e., 2-level registration with subgroups), "no clustering" analysis applied the IST-G tensor registration to all scans together as a single group (i.e., 2-level registration without subgroups), where the IST-G tensor registration involves two sequential applications of standard tensor-based registration (Fig. 1B and Fig. 2C).This rationale is now included on page 11, lines 253-254.

13. Can the authors explain how were the four metrics to evaluate registration performance computed? Were they extracted using the DTI-TK software? If not, the authors should specify this information in the text.

Response: Thank you for this question. Two (dyadic coherence and normalized standard deviation of FA) of the four metrics were computed using DTI-TK. The dyadic coherence was computed using “TVMean” command and the normalized standard deviation of FA was computed using “SVMean” command. Jacobian determinant was computed using “CreateJacobianDeterminantImage” command in Advanced Normalization Tools (ANTs). Normalized mutual information was computed using the Matlab package in 2. We now provide these details in the revised manuscript (see highlighted section, page 12, lines 276-284).

14. The quality of all main figures is poor. Can the author provide a better resolution? Also, it is recommended to avoid acronyms in the caption (or at least explain them first), allowing the reader to better interpret each figure.

Response: We apologize for this. We uploaded our figures to the journal with high resolution (300 dpi), but it appears that the journal PDF files have poorer quality. If it is possible to download the original figures within the submission (rather than viewing figures in the converted PDF), they should be of higher quality.

We now explain all acronyms before using them in figure captions.

15. Although the authors somehow provided details, it is still difficult to do not get lost when reading the different terminologies (e.g., intermediate subgroup common space, sample-specific common space, groupwise registration, standard registration) used in the text. If possible, can the authors provide more insights for these terms to avoid possible confusion?

Response: Thank you for this comment. We understand that the different terminologies may confuse readers. We now define those term

---

## [Editor Report · Decision Letter 1]

Groupwise registration of infant brain diffusion tensor images using intermediate subgroup templates

PONE-D-25-02132R1

Dear Dr. Shultz,

We’re pleased to inform you that your manuscript has been judged scientifically suitable for publication and will be formally accepted for publication once it meets all outstanding technical requirements.

Kind regards,

Federico Giove, PhD

Academic Editor

PLOS ONE
---

## [Editor Report · Acceptance letter]

PONE-D-25-02132R1

PLOS ONE

Dear Dr. Shultz,

I'm pleased to inform you that your manuscript has been deemed suitable for publication in PLOS ONE. Congratulations! Your manuscript is now being handed over to our production team.

Kind regards,

on behalf of

Dr. Federico Giove

Academic Editor

PLOS ONE